# YAP at the Crossroads of Biomechanics and Drug Resistance in Human Cancer

**DOI:** 10.3390/ijms241512491

**Published:** 2023-08-06

**Authors:** Miao Huang, Heyang Wang, Cole Mackey, Michael C. Chung, Juan Guan, Guangrong Zheng, Arkaprava Roy, Mingyi Xie, Christopher Vulpe, Xin Tang

**Affiliations:** 1Department of Mechanical and Aerospace Engineering, Herbert Wertheim College of Engineering, University of Florida, Gainesville, FL 32611, USA; 2UF Health Cancer Center, University of Florida, Gainesville, FL 32610, USA; 3Department of Biochemistry and Molecular Biology, University of Florida, Gainesville, FL 32603, USA; 4Department of Physics, University of Florida, Gainesville, FL 32611, USA; 5Department of Medicinal Chemistry, University of Florida, Gainesville, FL 32603, USA; 6Department of Biostatistics, University of Florida, Gainesville, FL 32603, USA; 7Department of Physiological Sciences, University of Florida, Gainesville, FL 32603, USA

**Keywords:** bioengineering, biomechanics, yes-associated protein (YAP), extracellular matrix (ECM), cancer, drug resistance, mechanobiology, mechanomedicine, CRISPR/Cas9 imaging

## Abstract

Biomechanical forces are of fundamental importance in biology, diseases, and medicine. Mechanobiology is an emerging interdisciplinary field that studies how biological mechanisms are regulated by biomechanical forces and how physical principles can be leveraged to innovate new therapeutic strategies. This article reviews state-of-the-art mechanobiology knowledge about the yes-associated protein (YAP), a key mechanosensitive protein, and its roles in the development of drug resistance in human cancer. Specifically, the article discusses three topics: how YAP is mechanically regulated in living cells; the molecular mechanobiology mechanisms by which YAP, along with other functional pathways, influences drug resistance of cancer cells (particularly lung cancer cells); and finally, how the mechanical regulation of YAP can influence drug resistance and vice versa. By integrating these topics, we present a unified framework that has the potential to bring theoretical insights into the design of novel mechanomedicines and advance next-generation cancer therapies to suppress tumor progression and metastasis.

## 1. Introduction

One of the most important lessons from cell biology in the past decades is that biomechanical forces, both endogenous and exogenous, are crucial in regulating the physiology and pathology of the human body. Living cells sense, transduce, and respond to specific biomechanical forces and other physical stimuli in their microenvironments [1,2,3,4,5,6], such as the geometry, dimensionality, porosity, and viscoelastic properties of the extracellular matrix. In turn, these biomechanical factors can fundamentally regulate and be altered by many diseases, including cancer, arthrosclerosis, heart disease, muscular dystrophy, and neurodegeneration [7,8,9,10,11]. Hence, understanding what the universal principles are that underpin these mechanobiological processes and how to mechanically modulate cell, tissue, and organ functions is a critical issue in science today.

In this review article, we will present a unified framework (Figure 1) to discuss the previously underappreciated interplay between a key mechanosensitive protein, YAP (yes-associated protein), and the development of drug resistance in human cancer cells (as detailed in Section 5). This new framework holds the potential to facilitate the creation of novel cellular engineering and therapies through the lens of biophysics and engineering. A deeper understanding of YAP-associated mechanotransduction processes in health and diseases will likely facilitate the next-generation of biomaterials, soft bio-robotics, wearable devices, and implants.

YAP and transcriptional co-activator with PDZ-binding motif (TAZ) are transcriptional co-activators that critically facilitate the signal transduction from biomechanical stimuli into gene expression [12,13,14]. In general, YAP is expressed in both the cell nucleus and cytoplasm. However, only when it is in the nucleus does YAP perform biological functions, mainly by binding to transcription factors including transcriptional enhancer activator domain (TEAD) to regulate cell proliferation, oncogenic transformation, epithelial-to-mesenchymal transition (EMT), and drug resistance [15,16]. Conventionally, YAP is considered the downstream of the Hippo signaling pathway, which is an evolutionarily conserved pathway to control organ size, development, and regeneration. In the human Hippo pathway, activation of Mst1/2 kinases sequentially phosphorylates Lats1/2 kinases and YAP. Activated Lats kinases phosphorylate YAP mainly at Ser127 site (other phosphorylation sites include S61, S109, S164, S381, T63, S138, S289, S351, and S384), promoting retention of YAP in the cytoplasm by binding to 14-3-3 proteins. Hence, pYAP-S127 level is widely used as an indicator of Hippo pathway activity [17,18]. However, pYAP-S127 alone does not determine the nuclear/cytoplasmic translocation of YAP and the downstream function. The detailed mechanism of YAP nuclear translocation will be discussed in Section 2.4.

In tumors, YAP plays important roles in regulating neoplasm initiation, growth, metastasis, and drug resistance. Importantly, overexpression and nuclear translocation of YAP occur in multiple tumor types [19] (Figure 2A). For example, in lung adenocarcinomas, 76% of the cases show overexpression of YAP and TAZ [19]. However, increasing evidence suggests that the genetic dysregulation of components within the Hippo pathway is rare (reviewed in ref. [12,20]). These data indicate that the abnormal YAP activities discovered in tumors are likely regulated by Hippo-independent mechanisms. Moreover, a dysregulated Hippo pathway may only induce dysregulated pYAP-S127. The combination of the Hippo pathway, Src-FAK pathway, and mechanical stimulus regulates the nuclear translocation of YAP. The overall nuclear translocation of YAP determines the change in the downstream gene expression and cellular function (Figure 2B,C).

Several pharmacological inhibitions of YAP or YAP-TEAD interaction show repression effects on tumor progression, although all current inhibitors are not clinically viable [19,42,43,44]. For example, Verteporfin binds with YAP and abrogates the YAP-TEAD interaction to reduce both tumor growth and drug resistance in multiple tumor types [42,45,46]. However, Verteporfin requires a high cellular concentration to effectively bind to YAP, which makes it currently not viable in clinical practices [42,43]. Poor target selectivity towards drug-resistant tumor cells is a major limitation of the pharmacological inhibition of YAP [19,43]. For example, the non-cell-specific inhibition of YAP worsens the fibrosis and vascular leakage of lung tissue in mice. However, targeting selectively expressed genes upstream of YAP inhibits lung fibrosis; Ref. [47] shows potential to partially overcome this challenge. Mechanistically, the DRD1 gene is selectively expressed in lung fibroblast and regulates YAP nuclear translocation. DRD1 agonism selectively inhibits YAP activity only in the DRD1-expressed fibroblast without affecting the viability of surrounding epithelial and endothelial cells, and thus reduces fibrosis [47]. These results indicate that pharmacological YAP inhibition will simultaneously reduce the survival of both drug-resistant cells and healthy normal cells. Instead, leveraging the unique characteristics of drug-resistant cells (e.g., selectively expressed genes or unique mechano-sensing properties) may overcome the limitation of selectivity. A similar strategy is also applicable in the inhibition of mechano-regulated YAP. Emerging evidence shows that tumor cells receive abnormal mechanical stimulus from surrounding tumor and tissue microenvironments, including elevated tissue stiffness, solid stress, and interstitial fluid shear forces. In addition, tumor cells show differential mechano-sensing compared to normal cells [48,49,50]. If drug-resistant cells (1) receive unique mechanical stimulus that can regulate YAP, and (2) show unique mechanism in mechano-regulated YAP different from that in the normal cells, then selectively targeting the unique stimulus or mechano-regulation mechanism in drug-resistant cells may reduce drug resistance without affecting normal cells.

In summary, YAP activity is critical in the drug resistance of cancer cells. Mechanical stimulus shows great potential in regulating YAP in cancer cells and reducing drug resistance (Figure 1). In the following sections, we will discuss three intertwined topics: (1) molecular mechanisms that underpin the mechanical regulation of YAP; (2) the contribution of YAP to drug resistance in cancer; and (3) the current understanding of reducing drug resistance using mechanically regulated YAP dynamics.

## 2. Mechanical Regulation of YAP Dynamics

In living cells, YAP is sensitive to diverse types of mechanical signals, reflected by its translocation between nucleus and cytoplasm (Figure 1). Cells constantly experience mechanical signals from the extracellular matrix (ECM), interstitial fluid, and neighboring cells [51,52,53,54]. Intracellular mechanosensors such as integrin, GPCR, mechanosensitive ion channels (Piezo ½ and Transient Receptor Potential (TRP) channels, and the cell nucleus can sense the extracellular mechanical signals to induce nuclear/cytoplasmic translocation of YAP [54,55,56,57]. Specifically, YAP can be influenced by three types of mechanical signals: (1) Mechanical properties of the ECM; (2) Actively applied extracellular force; and (3) Endogenous contractility and geometric attributes of the cell. These mechanical stimuli regulate YAP via different molecular mechanisms but most of the mechanisms involve nuclear mechano-sensing as the direct regulator of YAP. Next, we will discuss these three types of mechanical signals and their influences on YAP.

### 2.1. Viscoelastic Mechanics of ECM Regulates YAP

In general, ECM stiffness (Young’s modulus) is sensed by transmembrane integrins on the cell surface, which sequentially induce activation of Src, FAK at focal adhesion sites and intracellular Rho-ROCK machinery to increase cytoskeleton contractility, which is mechanical tension in cells generated by the sliding of actin and myosin filaments along each other [58,59,60,61]. As demonstrated by our own CRISPR/Cas9 imaging of real-time YAP dynamics and by other independent research groups, increased cytoskeleton contractility triggers nuclear translocation of YAP, while reduced contractility induces cytoplasmic retention of YAP [19,22,34,57,62,63,64]. For example, in human mammary epithelial cells, 80% of the cells cultured on stiff plastic (~10 MPa; 1 Pa = 1 Newton/m^2^) substrate show nuclear YAP translocation, while the rest 20% of the cells show homogenous intracellular YAP distribution. In contrast, the cells of the same type cultured on soft hydrogel (0.7 kPa) substrate do not show nuclear YAP translocation [22]. In adipose tissue-derived mesenchymal stem cells, the YAP nuclear/cytoplasm (N/C) ratio on 28 kPa substrate is 2-fold higher than the YAP N/C ratio on 1.5 kPa substrate [65]. Cytoskeleton contractility is necessary for substrate stiffness to regulate YAP; however, the exact mechanism of action is unknown [22,57].

Current understanding is that nucleus receives mechanical force from cytoskeletal contractility and induces conformational change in nuclear pores. Such conformational change regulates the active transport rate (both import and export) of YAP through nuclear pores to tune the overall dynamics of YAP nuclear/cytoplasmic translocation [57]. If cytoskeleton structure is disrupted or cytoskeletal contractility is inhibited, substrate stiffness sensed by integrin cannot be transmitted to the cell nucleus through tensed cytoskeletal filaments. Thus, nuclear pores do not change their conformation to regulate YAP active transport rate. However, exogeneous force applied on the cell nucleus can bypass the cytoskeletal force transmission between substrate and cell nucleus [57] and directly alter nuclear pores, which will be detailed in Section 2.2 and Section 2.3.

Viscoelasticity of ECM enhances nuclear translocation of YAP. For example, polysaccharide alginate gels with higher viscoelasticity (relaxation time = 30 s) induce nuclear YAP translocation in 10–50% of the cells, while gels with lower viscoelasticity (relaxation time = 350 s) induce nuclear YAP translocation in 0% of the cells. Recent evidence suggests that viscoelasticity regulates YAP through a FAK-Arp2/3-complex-dependent mechanism [66].

### 2.2. Geometric Attributes and Contractility of the Cell Regulate YAP

Cell spreading area (in 2D culture), i.e., area covered by a cell as it adheres on a 2D substrate, and cell volume (in 3D culture) positively regulate nuclear YAP translocation [22,57,65,67,68,69]. To differentiate the regulation of YAP by cell spreading area and focal adhesion area, recent studies have modulated focal adhesion area without changing cell spreading area. The data show that modulation of focal adhesion area results in no noticable regulation on YAP nuclear translocation [22]. However, inhibition of FAK activity induces decrease in YAP nuclear translocation [24]. These results indicate that, FAK activity rather than focal adhesion area regulates YAP nuclear translocation.

Cell contractility, cell spreading area and substrate stiffness often functionally intertwine with each other to influence dynamics of YAP and other cell signaling effectors [70,71]. positively regulates YAP nuclear translocation and shows temporal correlation with YAP, while inhibition of contractility suppresses nuclear translocation of YAP [22,57,71]. Contractility, however, is not absolutely necessary for YAP nuclear translocation if the force is directly applied to the nucleus [57]. These data indicate that contractility serves as a key bridge for the force transmission from cell membrane into the nucleus to mechanically regulate YAP.

### 2.3. Actively Applied Extracellular Forces Regulate YAP

Cells constantly receive mechanical tension and compression forces from the surrounding tissues and cells, especially in the lung and heart tissues. Both static and cyclic stretching of the cells induce nuclear YAP translocation [22,64,72,73]. Cytoskeleton structure rather than contractility is necessary for the stretching-induced YAP translocation [72,73]. Vertical compression (24 Pa) on HeLa and MCF-10A cells by polydimethylsiloxane (PDMS) causes F-actin depolymerization and suppresses YAP nuclear translocation [51]. In contrast, vertical compression (1.5 nN) by atomic force microscopy (AFM) induces YAP nuclear translocation with disrupted cytoskeleton [57]. These results indicate that, independent from the existence of cytoskeletal contractility, nuclear-force-sensing alone is sufficient to regulate YAP nuclear translocation.

### 2.4. Mechanism of Mechanically Regulated YAP Nuclear Translocation

In a high percentage of solid tumors, YAP/TAZ activities of tumor cells are deregulated, while the conventional Hippo pathway is not [12,19,20]. Because YAP/TAZ activities are mechano-sensitive and solid tumors present aberrant mechanical microenvironment to their constituent cells, it is natural to hypothesize that the deregulated YAP/TAZ activities observed in these tumors are due to the altered mechanical microenvironments in tumors but not the Hippo pathway. Hence, how mechanical stimulus regulates YAP nuclear translocation is under active investigations today.

Several research groups show that mechanical force regulates YAP nuclear translocation through (1) inducing unique phosphorylation of YAP, and (2) altering conformation of nuclear pores, competing with conventional Hippo-pathway-induced phosphorylation. For example, osmotic pressure increases pYAP-S127 indicating activation of Hippo pathway, and simultaneously induces Hippo-independent pYAP-S128 to disrupt the binding between pYAP-S127 and 14-3-3, eventually resulting in increased nuclear translocation of YAP [74]. Recent research shows that, independent from Hippo pathway, activation of FAK phosphorylates S397 site of YAP in mice (S381 in human) and Y357 site in human to promote YAP nuclear translocation [23,75,76]. In this study, mechanical force is not directly investigated in the FAK-induced YAP nuclear translocation. However, FAK activity is known to be sensitive to mechanical stimulus [27,28,29,30,31], indicating that mechanical stimulus potentially regulates both phosphorylation and nuclear translocation of YAP through enhanced FAK activity.

Besides phosphorylation of YAP, mechanical stimulus regulates YAP nuclear translocation through regulating nuclear pore conformation. Because YAP lacks nuclear localization sequence, it needs to bind with importin to enter the cell nucleus by active transport through the nuclear pore. Compressing of nucleus changes the conformation of nuclear pore and increases the importing rate of YAP into the nucleus [57,77]. However, phosphorylation of YAP other than pYAP-S127 is not investigated. One remaining question in the field is that whether force-induced conformational changes in nuclear pores regulate YAP purely through physical restriction, or is phosphorylation concurrently involved? Collectively, these datasets suggest that: (1) pYAP-S127 serves as an indicator of Hippo pathway activity alone; (2) phosphorylated YAP at other different sites together likely contribute to YAP nuclear translocation; and (3) Mechanical regulation of nuclear pore size simultaneously regulates YAP nuclear translocation.

In summary, YAP can be regulated by mechanical signals through nuclear mechano-sensing, either endogenous-contractility-mediated or exogenous-force-regulated. Therefore, we propose that possible mechanical strategies that aim to inhibit YAP activity may target (1) changing extracellular mechanical signals; (2) changing force transmission in cytoskeleton; and (3) changing nuclear mechano-sensing efficiency.

## 3. YAP and Drug Resistance

Drug resistance is one of the most prominent obstacles encountered during cancer treatment [78,79,80,81]. Today many available drugs in clinical practices can provide initial promising prevention of tumor progression. However, despite the high efficacy of these drugs, a subpopulation of tumor cells often survives the initial drug treatments, due to the pre-existing drug resistance or by developing capability of drug resistance during treatment, and re-grow into new tumors after the drug administrations. These drug-resistant tumor cells are likely the real culprit in causing cancer mortality [78,82]. Hence, to design next-generation comprehensive therapeutic strategies to eradicate cancer, elucidation of the mechanisms underpinning their drug resistance is one utmost goal in cancer research [83]. Interestingly, increasing recent evidence indicates that nuclear expression of mechano-sensitive YAP proteins contributes significantly to the increase in drug resistance in multiple types of cancer cells. In this section, we review the most current data and understanding of YAP’s contribution to drug resistance in lung cancer, as well as their mechanisms of action (Figure 2).

### 3.1. YAP Is More Activated in Drug-Resistant Cancer Cells

Recent studies indicate that YAP proteins tend to be more activated (i.e., YAP nuclear translocation) in drug-resistant cancer cells than in non-drug-resistant ones. The increased YAP nuclear translocation can induce anti-apoptotic and pro-proliferative phenotypes which can contribute to drug resistance and tumor relapse (Figure 2B,C). Based on the reported mechanisms that activate YAP, we have categorized these studies into two groups.

In one group, drug-resistant cancer cells, either immortalized cell lines or primary tumor cells, are found to be intrinsically YAP-rich intrinsically (i.e., high YAP expression in the cell nucleus) compared to that of non-drug-resistant cells. One hypothesis is that the intrinsic high YAP expression in these cancer cells results in their drug resistance capability. Another hypothesis is that drug treatment triggers YAP nuclear translocation in these cells thus increases the drug resistance. For instance, Lee TF et al. find that upregulated YAP expression in EGFR-mutant lung cancer cells led to the development of resistance against Epithelial Growth Factor Receptor (EGFR) tyrosine kinase inhibitors (TKIs) gefitinib, afatinib and osimertinib [84]. McGowan et al. find higher nuclear YAP concentration in an osimertinib (EGFR-TKI)-resistant H1975 lung cancer cell line in culture [85]. Song et al. find increased YAP expression and stemness in A549 lung cancer spheres compared with normal A549 adherent lung cancer cells, and note an association between increased YAP and enhanced cisplatin resistance [86]. Further, Kirsten Rat Sarcoma (KRAS) mutated tumor show resistance to Mitogen-Activated Protein Kinase Kinase (MEK) inhibitor monotherapy. Increased YAP nuclear translocation is observed in KRAS-mutated cancer cells [21,87]. For excample, Cheng H. et al. observe higher YAP1 nuclear staining in EGFR- or KRAS-mutated lung adenocarcinoma in comparison to EGFR/KRAS wild type [21]. Lin L. et al. observe increased nuclear YAP expression in v-Raf murine sarcoma viral oncogene homolog B1 (BRAF)-mutant lung cancer cells (specifically in tumors encoding BRAF V600E NSCLC and melanoma) and in KRAS-mutant NSCLC tumors [87]. Taken together, these findings suggest that the heightened activation of YAP, as observed in EGFR- or KRAS-mutated lung adenocarcinoma, as well as BRAF-mutant lung cancer cells and KRAS-mutant NSCLC tumors, might play a significant role in conferring resistance to MEK inhibition. Together, these studies reveal that drug-resistant lung cancer cells tend to have high YAP activation, suggesting an important correlation between YAP expression and drug resistance.

In the other group, drug treatment directly increases YAP nuclear translocation in a subpopulation of treated cancer cells and thus actively induce drug resistance. It is worth noting that this second group may overlap with the first one to a certain level. For instance, Kurppa KJ et al. find that YAP nuclear localization increased significantly in EGFR-mutant NSCLC after 10-day combined treatment with EGFR-TKIs osimertinib and trametinib [88]. Similarly, Yamazoe et al. find YAP nuclear localization is induced by short-term (72 h) lorlatinib treatment in ROS1-rearranged KTOR71 NSCLC cells and promote cell survival through activation of AKT. Meanwhile, the treated cells exhibit elongated morphology and increased cell-ECM adhesion signature in the proteome analysis [89].

In both groups, the studies propose various direct mechanisms by which YAP nuclear activation increases after drug treatment. For example, by monitoring LATS1 kinase, Kurppa KJ et al. find decreased LATS1 phosphorylation in response to drug treatment, suggesting YAP activation can be drug-treatment-induced by influencing Hippo pathway [88]. In exploring possible driver genes promoting YAP in drug-resistant lung cancer cell lines, Lee TF et al. detect an upregulation of Her2 mRNA expression [73]. Because knockdown of Her2 can reduce YAP expression, Her2 is hypothesized to have a role in increasing YAP activation and subsequent TKI-resistance. McGowan et al. note that erlotinib-, gefitinib- and osimertinib-resistant lung cancer cell lines have reduced expression of E-cadherin, a protein previously is found to promote YAP degradation [85]. Yamazoe et al. find that short-term (72 h) ROS1-TKI lorlatinib treatment in ROS1-rearranged NSCLC, shows no effect on Lats1 phosphorylation but increases YAP nuclear translocation [90]. These data indicate that lorlatinib regulates YAP in a Hippo-indenpend mechanism. Importantly, they also find that cellular exposure to lorlatinib upregulates proteins involved in cytoskeleton and morphology changes. Since it is known that larger cell spreading induces nuclear YAP translocation, these data suggest that lorlatinib-induced elongation in morphology induces YAP activation and nuclear localization, indicating the novel crosstalk between mechanobiological phenotypes and pharmacological actions. Future studies are needed to clarify the mechanisms that lead to increased YAP activation in various drug-resistant cell types, contributing knowledge which may be crucial in developing new therapeutic strategies.

### 3.2. Artificially Changed YAP Expression Directly Influences Drug Resistance

To explore a stronger causal link between YAP expression and drug resistance, several studies examine the impact of artificially overexpressing and inhibiting YAP on drug resistance in cancer cells. In general, these studies have found that increased YAP activation induces drug resistance in cancer cells. For instance, Kurppa KJ et al. find that re-expression of wild-type YAP1 in YAP1 KO EGFR-mutated NSCLC re-induces resistance to EGFR-TKIs osimertinib and trametinib [88]. Lee TF et al. find that both overexpression of wide type YAP and constitutional activated YAP-5SA promote drug resistance in HCC827 and H1975 lung cancer cells [84].

In a similar fashion, various studies find that decreased YAP activation causes drug-resistant cancer to become sensitive to drug treatment. For instance, Cheng H. et al. finds that knockdown of YAP by either shRNA or siRNA improves sensitivity to cisplatin treatment in PC9 lung cancer cells; to radiation treatment in PC9, HCC827 and H157 lung cancer cell lines; and to EGFR tyrosine kinase inhibitor (TKI) erlotinib in EGFR-mutated NSCLC PC9 and HCC827 cell lines [21]. They also find that combining YAP1 knockdown and platinum therapy induces apoptosis in PC9 lung cancer cells. Finally, they find that verteporfin, a pharmacological inhibitor of YAP-TEAD interaction, sensitizes PC9 lung cancer cells to cisplatin, radiation, and erlotinib treatment [21]. Kurppa KJ et al. find that YAP1 knockout induced accelerated apoptosis in EGFR-mutated lung cancer cells when treated with EGFR-TKIs osimertinib and trametinib [88]. Lee TF et al. find that dasatinib, a Src family kinase inhibitor, effectively reduces YAP expression levels and EGFR-TKI--resistance of HCC827 lung adenocarcinoma cells [73]. Furthermore, combined EGFR and YAP inhibition effectively reduced the viability of TKI-resistant cells. Lin L et al. find that shRNA-mediated YAP1 knockdown in HCC364 cells enhances sensitivity to vemurafenib and to MEK inhibitor trametinib [87]. McGowan et al. find significant increased sensitivity to EGFR-TKIs erlotinib and gefitinib, as well as to T790M-specific osimertinib, after YAP knockdown in drug-resistant lung cancer cell lines [85]. Song et al. find siRNA-mediated knockdown of YAP resensitized A549 cells and cell spheres to cisplatin treatment, suggesting reactivation of apoptotic cascades as a primary mechanism of resensitization [84]. Yamazoe et al. find that inhibition of YAP enhances sensitivity to ROS1-TKI lorlatinib and induces apoptosis in ROS1-rearranged KTOR71 NSCLC cells, and that combination therapy with YAP inhibitor verteporfin and lorlatinib suppresses tumor regrowth in vivo [84].

Together, the collective evidence strongly supports a causative relationship between YAP activation and drug resistance in cancer cells (Figure 2B,C). Moreover, the findings suggest that combined YAP inhibition and drug treatment hold promise as a potential therapeutic strategy to overcome drug resistance in clinical practices.

### 3.3. Mechanisms of YAP-Mediated Drug Resistance

Next, we review and discuss the various proposed mechanisms of YAP-mediated drug resistance.

In the upstream of YAP activation, drug treatments potentially influence the positive/negative upstream regulators of YAP, such as NF2, Src, and FAK. For example, NF2 inhibits YAP activation through Hippo pathway [12,14]. NF2 mutation contributes to drug resistance in multiple cancer types [89,90], while molecular alterations of NF2 is found in 5% of lung adenocarcinoma patients and 15% of lung squamous cell carcinoma patients [91]. Inhibition of NF2 increases the drug resistance of NSCLC and melanoma [89]. In addition, drug treatment induces activation of Src [92,93]. Src regulates YAP to induce EGFR-TKI resistance through three main mechanisms: (1) the direct phosphorylation; (2) the activation of pathways that repress Hippo kinases; and (3) Hippo-independent mechanisms (reviewed in ref. [94]). Importantly, because Src activation is related to FAK activation in mechano-sensing [54], we propose that mechanical regulation of FAK may be one of the Hippo-independent mechanisms of Src-YAP regulation.

In the downstream of YAP activation, nuclear translocation of YAP potentially contributes to drug resistance, through (1) transcriptional regulation of anti-apoptosis genes; and (2) crosstalk with MAPK pathway and PI3K-AKT pathway. First, in multiple types of human normal and cancer cells, YAP binds to transcriptional factor and upregulates the expression of Connective tissue growth factor (CTGF) and Cyr61 to increase drug resistance [15]. Mechanistically, in breast cancer cells, overexpression of CTGF upregulates anti-apoptotic factor Bcl-xL and cellular inhibitor of apoptosis protein 1 (cIAP1) to increase drug resistance [95]. In osteosarcoma, CTGF promotes drug resistance through upregulation of surviving expression and nuclear factor kappa B (NF-κB) pathway [96]. Enhanced Cyr61 expression increases drug resistance through upregulation of NF-κB-regulated anti-apoptotic gene XIAP [97].

In parallel with these upstream and downstream events, the activated YAP coordinates with multiple intracellular cytoplasmic and nuclear molecules to collectively regulate drug resistance. For example, Kurppa KJ et al. assert that YAP mediates the evasion of apoptosis by repressing the induction of the pro-apoptotic protein BMF (Bcl2 Modifying Factor) via engagement of Epithelial-Mesenchymal-Transition (EMT) transcription factor SLUG [88]. Lin L et al. postulate that YAP and RAF-MEK signaling work in parallel to enhance expression of BCL-xL in RAF- and MEK-mutant cells, thereby inducing drug resistance [87]. Song et al. propose anti-apoptosis as a primary mechanism of cisplatin-resistance in A549 lung cancer cells and cell spheres [86]. They also suggest that YAP transcriptional regulation of ABCB1 protein, a glycoprotein involved in multi-drug resistance, may be a potential mechanism of induced drug resistance. Yamazoe et al. propose that YAP1 mediates initial ROS1-rearranged NSCLC cell survival in response to ROS1-TKI lorlatinib through AKT signaling [38].

Together, these diverse studies highlight new potential pathways, proteins, and mechanisms outside of the known causal effects of YAP which may play a role in YAP-mediated drug resistance in lung cancer. However, prior to effectively targeting YAP to reduce drug resistance, we need to address two questions that remain poorly understood to date: (1) whether mechanically regulated and biochemically regulated YAP contribute to drug resistance in the same mechanism? The phosphorylation sites of YAP due to mechanical signals are different from YAP-S127 in Hippo pathway but its exact sites are unknown (only pYAP-S128 is identified in the osmotic-pressure-induced YAP nuclear tranlocation). Such structural and chemical differences may contribute to distinct binding of YAP with downstream transcription factors and differentially regulated cellular functions. An exact identification of these sites will facilitate the innovation of precise medicine; (2) after inhibition of YAP, how the activation of compensating or redundant pathways (e.g., MAPK and PI3K-AKT pathway) reduces the drug resistance? A deeper understanding of their functional dynamics will promote the design of combinatorial strategies that may be necessary to increase the efficacy of YAP inhibition.

### 3.4. YAP and Immunotherapy

Emerging studies have shown that YAP (yes-associated protein) likely has a significant role in regulating the cellular response to immunotherapy in non-small cell lung cancer (NSCLC). Specifically, YAP has been found to regulate the expression of Programmed Death-Ligand 1 (PD-L1), which serves as a predictive biomarker for anti-PD-1/PD-L1 immunotherapy.

The binding of PD-L1 with its receptor, PD-1, promotes T-cell tolerance and enables tumor cells to evade immune surveillance. To counteract this undesired behavior, one effective therapeutic method developed has been the anti-PD-1/PD-L1 immunotherapy. The level of PD-L1 expression in a tumor can provide information about its likelihood to respond to anti-PD-1/PD-L1 immunotherapy treatment: Typically, tumors with high PD-L1 expression are more effectively treated with anti-PD-1/PD-L1 immunotherapy, whereas tumors with low or no PD-L1 expression have a lower effectiveness of response to these treatments.

However, despite the success of anti-PD-1/PD-L1 immunotherapy in clinical, a considerable number of tumors do not respond to PD-1/PD-L1 inhibitors even though they possess high PD-L1 expression. This clinical challenge highlights the urgent need to understand the underlying mechanisms that contribute to this resistance. Several studies have shed light on the involvement of YAP in resistance to immunotherapy and its relation to PD-L1 expression. Hsu et al. find that, in H2052 and 211H human malignant pleural mesothelioma (MPM) cells, YAP binds to PD-L1 enhancer to promote expression of PD-L1. YAP expression positively regulates PD-L1 expression, examined by both overexpression and verteporfin-inhibition of YAP [98]. However, whether nuclear YAP translocation shows any causal regulation on PD-L1 expression is not investigated [67]. Additional studies contribute further evidence of YAP’s role in regulating PD-L1 expression, while also demonstrating how the YAP/PD-L1 regulatory relationship produces immunotherapy-resistant phenotypes. A study by Li L. et al. investigates the mechanisms by which Long-chain Acyl-CoA dehydrogenase (ACADL), an enzyme which has been found to act as a tumor suppressor in cancers, inhibits proliferation and enhances chemotherapeutic drug-induced apoptosis in lung adenocarcinoma [99]. The study reveals that ACADL prevents tumor immune evasion by inhibiting YAP and subsequently suppressing PD-L1 expression. Similar findings emerge in a study by Yu M. et al., in which researchers explore the mechanisms by which interferon-γ (IFN-γ) produces immunotherapy-resistant phenotypes in cancer cells. They find that IFN-γ promotes nuclear translocation and phase separation of YAP after anti-PD-1 therapy in tumor cells, indicating that YAP may be a primary mediator of the pro-tumor effect induced by IFN-γ [100].

Together, these findings suggest that YAP nuclear localization and YAP-induced increase in PD-L1 expression contribute to immune evasion, immunotherapy resistance, and drug resistance in lung adenocarcinomas.

## 4. Roles of ECM Played in Regulating YAP and Drug Resistance in Cancer Cells

Besides providing mechanical supports for tissue cells to reside in, extracellular matrix (ECM) functionally regulates physical-chemical interactions between cells and their surrounding microenvironments by influencing various mechanical stimulations, including substrate stiffness (both elastic and viscous properties), patterns of fluid shear stress, or active pressure [40,101,102,103,104,105]. These mechanical stimulations, along with biochemical cues, critically regulate the prognosis and behavior of cancers (Figure 3). Hence, the full investigation of the mechanistic roles that ECM has in cancers, especially in chemotherapy resistance, is critical for clinical practice.

### 4.1. ECM Induces YAP Nuclear Translocation and Influences the Resistance/Sensitivity of Cancer Cells to Chemotherapies

Increasing studies have demonstrated that YAP has mechanotransducive property which can be regulated by ECM mechanics [65]. Meanwhile, as discussed in Section 3, YAP’s expression has been proven to be closely associated with the regulation of drug resistance in multiple human cancer cell lines [12]. Therefore, a deeper understanding of how mechanical cues from the ECM induce resistance or sensitivity of cancer cells to chemotherapies via YAP’s activation and expression will offer promise for innovating new therapeutic methods from a mechanobiology perspective. In this section, we introduce and discuss the state of knowledge on the ECM-regulated chemoresistance and sensitivity of distinct cancer cell types (Figure 3).

A biphasic relationship between drug resistance and their ECM with various stiffness in breast cancer cells. Within 10 kPa (soft), intermediate stiff (38 kPa), and stiff (57 kPa) substrate, intermediate substrate rigidity (38 kPa) induces the highest level of integrin-linked kinase (ILK), nuclear YAP translocation and drug resistance [34].

Negative correlation between ECM stiffness and drug resistance is found in ovarian cancer cells (OCC). YAP has the highest nucleus/cytoplasm ratio (N/C ratio) at stiff substrate (25 kPa). However, OCC on soft substrate stiffness (0.5 kPa) shows higher resistance to cisplatin and paclitaxel, which are drugs widely used in chemotherapies of epithelial ovarian cancer. Whether the drug resistance is regulated by YAP remains to be investigated [35].

Positive correlation between ECM stiffness and drug resistance is found in two groups. First, Gao et al. found that hepatocellular carcinomas cells, which are sensitive to sorafenib (Sora-S cells), gained drug resistance when cultured on stiff substrate (4 kPa). Inversely, cells that have resistance to sorafenib (Sora-R cells) regain the sensitivity of sorafenib after adapting to the soft substrate (0.7 kPa). More importantly, the stiffness-induced YAP activation is confirmed to regulate cells’ mechano-adaption of chemotherapy. In experiments, by quantifying YAP in the nucleus and cytoplasm, researchers find that, first, YAP has higher nuclear localization in Sora-R cells than Sora-S cells; second, in Sora-R cells, the YAP nuclear localization on 4-kPa substrate is higher than that on the 0.7-kPa substrate [36]. After silencing YAP in Sora-R cells, the mechano-adaption ability of Sora-R cells disappeared, suggesting that YAP has functional roles in mechano-regulated chemoresistance of hepatocellular carcinoma cells [36]. Second, in HER2-amplified human breast cancer HCC1569 cells, both YAP nuclear translocation and drug resistance to lapatinib are higher on tissue culture plastic (2 GPa) than on Matrigel (400 Pa). In vitro knockdown of both YAP and TAZ successfully restricted the modulus-dependent resistance of lapatinib in in HCC1569 cells. Knockdown of YAP in HCC1569 cells induced a smaller tumor volume inside of mice, in contrast to that of the tumors induced by original HCC1569 cells [106].

Overall, no consistent relationship is found between (1) ECM stiffness and YAP; and (2) YAP and drug resistance. When targeting the drug resistance through mechanically regulated YAP, the strategies will be cancer-type- and drug-dependent. In addition to the observations that the changes in ECM’s properties can affect cancer cells’ chemotherapies, recent studies suggest that chemotherapies can change the ECM mechanics and contribute to drug resistance reciprocally. In BRAF-mutant melanoma, BRAF inhibitors (PLX4032) remarkably can remodel surrounding ECM by activating melanoma-associated fibroblasts (MAFs) through phosphorylation of MLC2/MYL9, which is the key regulator of actomyosin contractility. Activated MAFs are sufficient to produce denser collagen fibrils in the ECM and drive drug resistance to PLX4032 in BRAF-mutant melanoma cells. Remodeled ECM drives ERK activation and cancer cells’ resistance through integrin β1 and FAK signaling pathways [39]. Although YAP activity is not investigated in this research, the regulatory role of integrin and FAK on YAP indicates that YAP may be responsible for the integrin β1-FAK-induced drug resistance [23,24,25]. Further, in the PLX4032-resistant melanoma cancer cells, YAP activation is necessary for the maintenance of drug resistance. PLX4032-induced increasing of actin stress fibers is necessary for the YAP activation [37]. Increasing of actin stress fibers can facilitate the mechano-sensing of the cells to altered ECM stiffness [58,59,60,61] and may further activate YAP [54,55,56,57]. These results demonstrate a closed feedback loop for ECM-chemotherapy to influence each other in BRAF-mutant melanoma cancer cells mutually.

### 4.2. ECM Influences Resistance/Sensitivity of Cancer Cells to Chemotherapies in a YAP-Independent Manner

As discussed in Section 4.1, YAP is critical in mediating ECM-guided mechanotransduction to induce drug resistance in cancers. Importantly, besides YAP, other molecular effectors are also able to perform mechanotransduction to induce the chemotherapy resistance of carcinoma cells. Therefore, YAP likely is not the only molecular effector that links ECM mechanics to chemoresistance in cancer cells, indicating that the mechano-induced drug resistance of cancer can be YAP-independent (Figure 3).

For example, the stiffness of ECM influences cells’ sensitivity and reduces the efficiency of radiotherapies and chemotherapies by impairing DNA double-strand breaks (DSBs)’ repair [107]. This process is mediated through MAP4K4/6/7 kinases instead of LATS1/2 and YAP. The repair process of DSB has a deficiency in breast cancer cells cultured on substrates of low stiffness (0.5 and 1 kPa), which increases cellular sensitivity to genotoxic agents. The low stiffness inhibits DSB repair through modulating RNF8-mediated ubiquitin signaling. The RAP2-Hippo pathway (as an intracellular mechano-transductor) is a direct link between ECM stiffness and intracellular processes. Researchers further proved that Rap2 could be activated at low stiffness. Meanwhile, the downstream Hippo kinase MST1/2 and MAP4K4/6/7 kinases were activated at low stiffness conditions. Furthermore, researchers knocked out Rap2 and MST1/2 and MAP4K4/6/7 kinases. In experiments, cells with Rap2 knocked out become more resistant to ionizing radiation at low stiffness [107]. The knock out of MAP4K4/6/7 restores the survival rate of cells at low stiffness after applying irradiation. In conclusion, researchers showed that Rap2 and the downstream Hippo kinase MAP4K4/6/7, but not MST1/2, are required in the inhibition of DSB repair [107].

As a mechanosensitive transcriptional coactivator [108], YAP can be activated by changes in ECM and, in turn, induces cancer cells’ chemoresistance in many cases. However, besides YAP, the change in ECM’s properties can mediate drug resistance through other molecules, such as β 1 integrin signaling pathway with its downstream effectors [109]. As the rigidity of ECM increases, triple-negative breast cancer cells show increased resistance to sorafenib. Researchers found out that breast cancer cells have significantly higher drug resistance in the stiff substrate group (400 kPa Young’s modulus gel) than in the 50 kPa group. Meanwhile, two of the downstream effectors of β1 integrin, JNK, and P38, have higher activity and phosphorylation levels in cells cultured on higher stiffness substrates. Thus, JNK and P38 become potential candidates to explain stiffness-induced drug resistance [109]. To figure out the core mechanism that drives the mechano-induced sorafenib resistance, researchers further test ERK1/2′s activity and prove the higher phosphorylation of ERK on stiff substrates at early points after sorafenib was applied. As potential molecule candidates, inhibitors of JNK, p38, and ERK are applied, respectively, combined with sorafenib. Based on these experiments, the inhibition of JNK increased the efficiency of sorafenib and restricted the impact caused by substrate stiffness [109].

More studies have been conducted on integrin and the following signaling pathway. As Wang et al. discuss, the integrin β1/FAK/ERK1/2/NF-κB signaling pathway is activated in liver cancer cells under high-rigidity substrate conditions (16 kPa) [110]. Importantly, integrin β1 was proven to drive liver cancer cells’ proliferation. In the experiments, silenced integrin β1 inhibits the proliferation effects of SMMC-7721 and HepG2 cells. Hence, researchers apply the combination of integrin inhibitor (GLPG0187) with chemotherapy agents ADM/DDP on hepatocellular cancer xenografts to investigate potential therapies. Results show that both the inhibitor-only group and the group that contains chemotherapy agents and the inhibitor have obvious effects on suppressing tumor growth and prolonging mice survival time [110]. Intriguingly, integrin signaling can be a potential therapeutic target to improve anti-cancer drugs’ efficiency. In all these studies, the roles of YAP/TAZ in mediating drug resistance remain to be investigated.

Besides the Hippo signaling pathway and YAP, the c-Met/PI3K/Akt signaling axis and a mechanosensitive microRNA, miR-199a-3p, were also proven to have critical roles in the mediation of chemoresistance in ovarian cancer cells [111]. Researchers show that miR-199a-3p’s expression is negatively correlated with the chemoresistance of ovarian cancer and the clinical treatment outcome of cisplatin, paclitaxel, and platinum [37]. In experiments, cisplatin-resistant A2780 cells show significantly lower expression of miR-1990a-3p than cisplatin-sensitive A2870 cells. More importantly, the ectopic miR-199a-3p could make cisplatin resistance cell line A2780/DDP regain the sensitivity to cisplatin and make SKOV-3 cells sensitive to paclitaxel. Hence, the medication role of miR-1990a-3p in ovarian cancer cells was proven. Furthermore, in the patient’s tumor samples, the platinum-resistance tumor shows lower expression of miR-1993-3p than the platinum-sensitive tumor, which is consistent with the results from previous experiments. In the tumor microenvironment, mechanical stimulations other than substrate stiffness play a significant role in cancer cells’ progression and potential therapies. Hassan et al.’s study [112] critically indicates that the microfluid shear stress could downregulate the expression of miR-199a-3p through the c-Met/PI3K/Akt signaling pathway. As the miR-199a-3p is downregulated by shear stress under 0.02 dynes/cm^2^ ascitic shear stress, tumors could become more resistant to chemotherapies. Hence, these results indicate potential therapeutic targets in ovarian cancers.

More effectors that conduct the YAP-independent mechanotransduction need to be investigated and discussed [112]. Researchers examined ductal carcinoma in situ (DCIS) from patients’ samples and created a 3D culture model to mimic the ECM stiffness. In the patients’ sample group, YAP is inactivated at the early stage of cancer. This observation is essential because the increased stiffness of ECM could drive cancer cell invasion at the early stage. Therefore, the lack of YAP activity indicates that the changes in ECM stiffness could drive cancer cells’ invasion without YAP. Furthermore, cancer cells cultured on the 3D model show no YAP activities, as stiffness increased in the in vitro group [112]. Overall, the lack of YAP’s activity was proven in the mechanotransduction of breast cancer in both 3D culture and patient sample condition, which means this mechanotransduction is independent of YAP. This research indicates the possibility that the mechanotransduction from ECM to cancer cells, which induces cancer cells’ drug resistance, can be conducted through various mechanisms or channels other than YAP.

## 5. Conclusions and Outlook

YAP promotes drug resistance in cancer cells and can be regulated mechanically (Figure 1). Therefore, besides using pharmacological inhibition of YAP to suppress drug resistance, the mechanical inhibition of YAP has unique potential for the development of new cancer therapies from a new perspective. We hereby propose a unified framework that elucidates how the mechanically, biochemically, and pharmacologically regulated activation of YAP contributes to drug resistance in cancer (Figure 1). Importantly, our framework, for the first time, proposes the interplay between mechanical and pharmacological regulation, which lays down a new mechanobiology foundation for the development of new combinatorial therapies.

First, besides the Hippo pathway, YAP activation can be regulated mechanically. The mechanical stimuli include: (1) extracellular mechanical signals such as ECM viscoelastic properties, solid stress generated in tissue/tumor microenvironments, and compressive/tensile force applied by body motions; and (2) intracellular mechanical signals such as endogenous cellular contractility and nuclear mechanical properties. These mechanical stimuli regulate the activation of YAP (as detailed in Section 2) and result in drug resistance in cancer cells (as detailed in Section 3). Second, drug treatment can potentially activate the upstream mechanical regulators of YAP to activate YAP and induce drug resistance, in addition to the direct regulation of YAP without any crosstalk with mechanotransduction pathways (as detailed in Section 4). This interplay can be achieved through multiple avenues: (1) drug treatment can increase extracellular ECM stiffness [95] and solid stress [113]; (2) drug treatment can induce cytoskeleton remodeling [16,37,114,115], increase cellular contractility [116], and change nuclear mechanical properties [117]; and (3) drug treatment can tune the mechano-sensitivity of cells to magnify the influences of given mechanical stimuli [117]. These intricate mechano-pharmacological interactions have not been exploited in the field. Overall, this framework suggests that, besides pharmacological treatments, mechanical treatments of cancer cells, either through inhibiting the mechanical regulators of YAP or suppressing the drug-treatment-induced activation of mechanotransduction events, can potentially reduce drug resistance.

Based on this notion, we propose three types of methods for the mechanical inhibition of YAP and drug resistance in cancer cells: (1) pharmacological modulation of abnormal ECM mechanical properties in the tumors to suppress the mechano-induced YAP activation; (2) actively applying mechanical signals (e.g., ultrasound [118,119] and physical stretching [120]) at the tumor site to inhibit YAP activity with high spatial-temporal resolution and cell-type selectivity; and (3) if specific mechano-sensing pathways in cancer cells are known, disrupting the key mediators in the pathways to suppress YAP activation or to reduce YAP’s mechano-sensitivity to minimize drug resistance. To map the real-time interplay between mechanical forces and intracellular YAP-associated signaling dynamics, the high-resolution functional and structural imaging of Ca^2+^ signals and CRISPR/Cas9-endogenous-tagged proteins (e.g., YAP, Piezo ½, actomyosin machinery, and nuclear constituents), accompanied by mechanobiological characterizations using molecular tension biosensors and live-cell traction force microscopy, will be instrumental.

We envision that the present unified framework (Figure 1) will help the scientific community better understand the potential of mechanobiology and biophysics for biomedical research, particularly cancer medicine. By integrating mechanobiology and biomechanics principles with oncology clinical trials, there is reason to be optimistic that more creative multidisciplinary strategies that leverage mechano-medicine will be developed to prevent, detect, treat, and ultimately cure cancer.

## Figures and Tables

**Figure 1 ijms-24-12491-f001:**
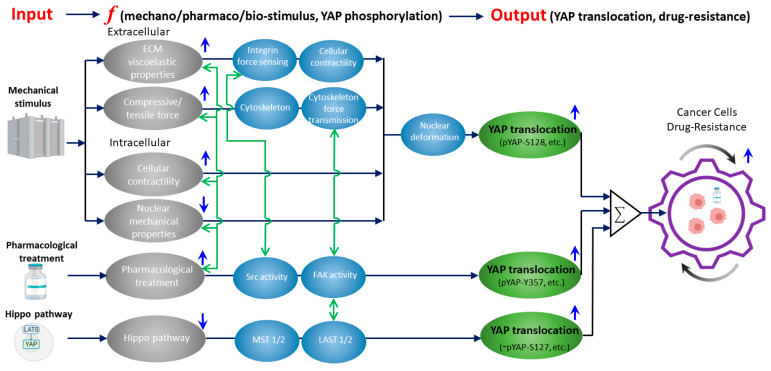
A unified framework that overviews the current knowledge of how mechanically, biochemically (Hippo), and pharmacologically regulated YAP activities contribute to drug resistance in cancer cells. Gray ellipses represent the input signals from mechanical (as detailed in Section 2), biochemical (Hippo), and pharmacological (as detailed in Section 3) upstream regulators of YAP. Blue arrows indicate whether the activation/inhibition of the upstream regulators will induce YAP nuclear translocation. Activation of mechanical stimuli (except nuclear mechanical properties) and pharmacological treatment upregulates YAP nuclear translocation. While inhibition of Hippo pathway upregulates YAP nuclear translocation. Black arrows show the corresponding YAP-regulatory pathways with the key molecular/cellular mediators listed in blue ellipses. Green ellipses represent the resulting YAP nuclear translocation for each pathway with the corresponding phosphorylation sites of YAP. In the Hippo pathway, YAP-S127 dephosphorylation is denoted as “−” (i.e., dephosphorylation). Green arrows show the current knowledge of the crosstalk between the inputs and key mediators (as detailed in Section 4 and Section 5). The crosstalk network indicates the potential targets of (1) inhibiting the mechanical regulators of YAP or (2) suppressing the drug-treatment-induced activation of mechanotransduction events to eventually reduce drug resistance (as detailed in Section 5).

**Figure 2 ijms-24-12491-f002:**
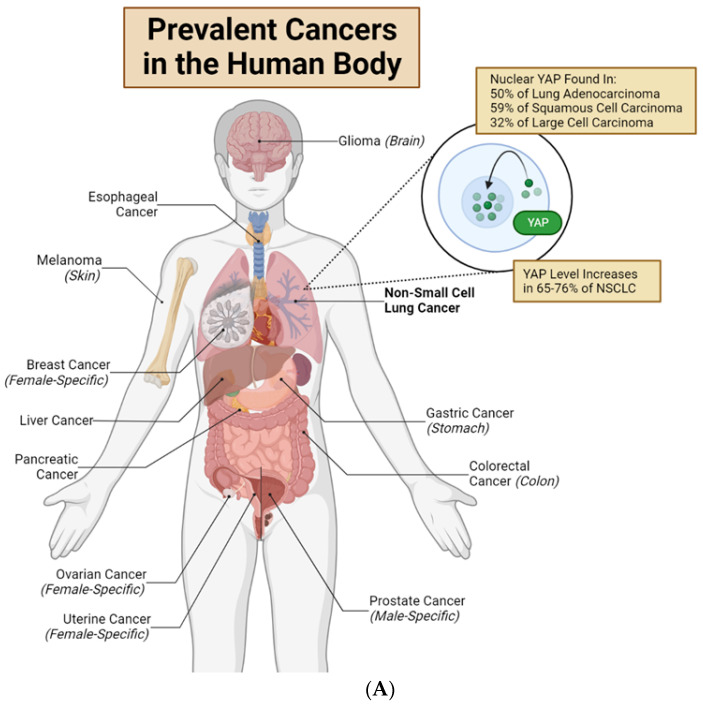
Nuclear YAP contributes to drug resistance of human cancer cells. (**A**) Major organs and associated common cancers in the human body. Non-small cell lung cancer (NSCLC) is bolded to highlight this review’s unique focus on YAP in lung adenocarcinomas and non-small cell lung cancer [12,21]. (**B**) Lung cancer cells with high cytoplasmic (low nuclear) YAP are drug-sensitive. EGFR-TKIs, RAF inhibitors, and MEK inhibitors target mutated EGFR, RAF or MEK proteins, respectively. This causes loss of anti-apoptotic and pro-proliferative downstream effects, resulting in drug-induced cell death. Cisplatin and radiation directly target cancer cell DNA, inducing cell death. (**C**) Lung cancer cells with high nuclear (low cytoplasmic) YAP are often drug-resistant. YAP in the nucleus will bind to TEAD transcription factors to promote transcription of genes which produce anti-apoptotic/pro-proliferative/senescence-inducing proteins and pathways. These downstream mechanisms lead to drug resistance, anti-apoptosis, and senescence. New research shows that several Hippo-independent pathways can regulate YAP activity (Figure 3). First, biophysical stimulus, such as tissue stiffness, regulates the nuclear translocation of YAP and serves as an “on/off” switch of the Hippo pathway to regulate YAP in parallel [22]. The mechanism of this mechanically regulated YAP dynamics will be explained in detail in Section 2. Second, a few biochemical signals can regulate YAP, independent of the Hippo pathway, through focal adhesion kinase (FAK) [23,24,25] and Aurora A kinase [26]. However, FAK activity is downstream of cells’ integrin-mediated sensing of biophysical signals [27,28,29,30,31]; therefore, whether the FAK-YAP pathway is part of the mechano-regulated YAP pathway is currently under active investigation. T-bar arrows indicate drug-induced inhibition; black arrows indicate pathway activation, stimulation or promotion; and a red ‘X’ over an arrow denotes pathway loss. Created with Biorender.com.

**Figure 3 ijms-24-12491-f003:**
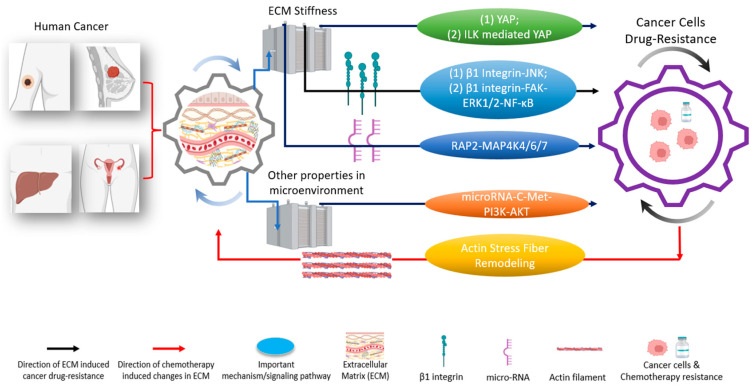
The tumor microenvironment and extracellular matrix (ECM) induce different pathways to regulate drug resistances in various types of cancer [19,32]. These pathways are regulated through either YAP or other critical mechanisms [33]. Biophysical-chemical changes in ECM’s properties, such as cells’ substrate stiffness, could drive cancer cells’ resistance (black arrows) against chemotherapies through YAP and/or YAP-related pathways (green; two representative pathways (1) and (2) are indicated), integrin (blue; two representative pathways (1) and (2) are indicated), and other signaling pathway instead of YAP (orange) [34,35,36,37]. Besides ECM stiffness, other signals in the tumor microenvironment, such as shear stress, can induce cancer cells’ drug resistance (brown) [38]. Conversely, the application of chemotherapies in cancer cells could cause changes in the ECM’s properties (red arrow) [39]. Understanding the functional networks between ECM, ECM’s mechanical stimulations, and cancer drug resistance opens the door to designing and developing next-generation effective clinical treatments [15,40,41]. Created with Biorender.com.

## Data Availability

Publicly available datasets were analyzed in this study. This data can be found in the references cited.

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
