# Peer review of "YAP at the Crossroads of Biomechanics and Drug Resistance in Human Cancer"

_ijms, 2023, doi:10.3390/ijms241512491_

Round 1

Reviewer 1 Report

Title: YAP at the Crossroads of Biomechanics and Drug Resistance in Human Cancer

Journal: IJMS

Manuscript ID: ijms-2534418

General opinion

The authors systematically sorted out the characteristics and mechanisms of YAP in biomechanics and drug resistance in human cancer, which can provide a theoretical basis for the development of new tumor treatment methods. The explanation of this paper is logical and reasonable, and the diagram is clear, but the expression of the proposed unified framework is not clear, and the description is not detailed enough. Therefore, major revisions to this review are recommended. Comments are as follows:

Major comments

1.    The abstract mentions "present a unified framework", but what unified framework refers to is not found below, and a concluding statement can be added appropriately.

2.    The part of "YAP and Drug Resistance" only systematically lists the existing research results, lacks its own views, and suggests combining the mechanism of drug resistance and the regulation mechanism of YAP to put forward innovative views and make greater contributions to later research.

3.    Section 4 of the article focuses on the role of ECM and has been mentioned several times before, so it is recommended to add it to the keyword section.In addition, the keywords are somewhat trivial and need to be reduced, such as mechanical regulation; mechanobiology; mechanomedicine.

Minor comments

1.    Please double-check the format of the references, keep the spacing consistent, remove extra spaces, and punctuate in English. For example: page 17, lines 660, 670, 676-680, page 21, lines 905, 916.

2.    Subheading 4.2 (page 9, line 439) does not require colons and needs to be consistent with other headings.

3.    It is proposed to modify 0% of the expression (page 3, lines 135,138)

Minor editing of English language required

Reviewer 2 Report

In this review, Huang and co-workers discuss the emergence of YAP/TAZ as mechanosensitive controllers of tumor growth and aim to provide a conceptually tight framework for the development of novel anti-tumor drugs based on YAP/TAZ targeting independent of the Hippo pathway. Overall, this is a timely and interesting topic, but the review is largely incomplete, thus I cannot recommend publication in its present form, although I’d be very interested in a revised version that addresses the following:

1)      As it is, the review is largely phenomenological, describing things that happen, but not discussing why. While a review that agglutinates the appropriate references is a useful resource, the authors are supposed to be experts in the field, hence their opinions on some of the whys would be very interesting for the community. Some examples: YAP in plastic is nuclear in 80% of the cells (lines 134-135): what is different about the remaining 20%? Cytoskeletal contraction is needed, why? Can it be bypassed with external forces? Nuclear compression?

2)      A general section on YAP regulation (including phosphorylation!) and mechanisms of nuclear translocation (work from Roca-Cusasch and others) is mandatory.

3)      Since YAP/TAZ are transcription factors, a thorough discussion of the YAP-dependent proteome involved in drug resistance is required.

4)      The authors float an important idea that is not really developed, that YAP/TAZ appear deregulated in a high percentage of tumors, yet the Hippo pathway is not, hence the source of the deregulation is not aberrant Hippo signaling but possibly mechanical alterations. This is underdeveloped and a major source of interest.

English is fine and only typesetting errors need to be checked. Beyond my purview.

Round 2

Reviewer 1 Report

No further suggestions, agreed to publish

No further suggestions, agreed to publish

Reviewer 2 Report

Thanks for taking my comments into consideration. Good job. Happy to recommend acceptance.

Fine